# Crystal structure of a multi-domain human smoothened receptor in complex with a super stabilizing ligand

Xianjun Zhang[1,2,3,4], Fei Zhao[1], Yiran Wu[1], Jun Yang[5], Gye Won Han[6], Suwen Zhao[1,2], Andrii Ishchenko[6], Lintao Ye[1,2,7], Xi Lin[1,2,3,4], Kang Ding[1,2,8], Venkatasubramanian Dharmarajan[9], Patrick R. Griffin[9], Cornelius Gati[10], Garrett Nelson[11], Mark S. Hunter[12], Michael A. Hanson[13], Vadim Cherezov[6], Raymond C. Stevens[1,2], Wenfu Tan[5], Houchao Tao[1] & Fei Xu[1,2]

The Smoothened receptor (SMO) belongs to the Class Frizzled of the G protein-coupled receptor (GPCR) superfamily, constituting a key component of the Hedgehog signalling pathway. Here we report the crystal structure of the multi-domain human SMO, bound and stabilized by a designed tool ligand TC114, using an X-ray free-electron laser source at 2.9 Å. The structure reveals a precise arrangement of three distinct domains: a seven-transmembrane helices domain (TMD), a hinge domain (HD) and an intact extracellular cysteine-rich domain (CRD). This architecture enables allosteric interactions between the domains that are important for ligand recognition and receptor activation. By combining the structural data, molecular dynamics simulation, and hydrogen-deuterium-exchange analysis, we demonstrate that transmembrane helix VI, extracellular loop 3 and the HD play a central role in transmitting the signal employing a unique GPCR activation mechanism, distinct from other multi-domain GPCRs.

[1] iHuman Institute, ShanghaiTech University, 2F Building 6, 99 Haike Road, Pudong New District, Shanghai 201210, China. [2] School of Life Science and Technology, ShanghaiTech University, Shanghai 201210, China. [3] Institute of Biochemistry and Cell Biology, Shanghai Institutes for Biological Sciences, Chinese Academy of Sciences, Shanghai 200031, China. [4] University of Chinese Academy of Sciences, Beijing 100049, China. [5] Department of Pharmacology, School of Pharmacy, Fudan University, Shanghai 201203, China. [6] Departments of Chemistry, Biological Sciences and Physics & Astronomy, Bridge Institute, University of Southern California, Los Angeles, California 90089, USA. [7] Shanghai Institute of Materia Medica, Chinese Academy of Sciences, University of Chinese Academy of Sciences, 555 Zuchongzhi Lu, Building 3, Room 426, Shanghai 201203, China. [8] Key Laboratory of Computational Biology, CAS-MPG Partner Institute for Computational Biology, Shanghai Institutes for Biological Sciences, University of Chinese Academy of Sciences, Chinese Academy of Sciences, Shanghai 200031, China. [9] Department of Molecular Therapeutics, The Scripps Research Institute, 130 Scripps Way, Jupiter, Florida 33458, USA. [10] Medical Research Council, Laboratory of Molecular Biology, Cambridge, Biomedical Campus, Francis Crick Avenue, Cambridge CB2 OQH, UK. [11] Department of Physics, Arizona State University, Tempe, Arizona 85287, USA. [12] Linac Coherent Light Source, SLAC National Accelerator Laboratory, 2575 Sand Hill Road, Menlo Park, California 94025, USA. [13] GPCR Consortium, San Marcos, California 92078, USA. Correspondence and requests for materials should be addressed to W.T. (email: wftan@fudan.edu.cn) or to H.T. (email: taohch@shanghaitech.edu.cn) or to F.X. (email: xufei@shanghaitech.edu.cn).

The Hedgehog (Hh) signalling pathway plays a key role in embryonic development and the regulation of adult stem cells. Uncontrolled activation of the Hh pathway results in numerous cancers in the brain, muscle and skin, and has drawn extensive attention from the drug discovery perspective[1]. The smoothened receptor (SMO)[2,3], a Class Frizzled seven-transmembrane helices (7TM) G protein-coupled receptor (GPCR), is a key component in this signalling pathway[4]. The activity of SMO is suppressed by the PTCH receptor[1], a 12TM protein. This suppression is disabled when Hh binds to PTCH, leading to phosphorylation of SMO's cytoplasmic region[5], which induces the translocation of GLI transcription factors into the nucleus to activate target genes[6]. However, the interaction between PTCH and SMO, and the release of PTCH suppression by Hh binding are not clearly understood. Previous biochemical and functional characterization studies have indicated that SMO contains at least two non-overlapping ligand binding pockets[7]. One of them is located inside the transmembrane domain (TMD) resembling the canonical ligand binding pocket in Class A GPCRs, targeted by numerous small molecules, including inhibitors and activators[3,8,9]. Another ligand-binding site is situated on the surface of the extracellular cysteine-rich domain (CRD), targeted by 20(S)-hydroxycholesterol (OHC) and other cholesterol analogues[10–12]. SMO constructs without the CRD show increased constitutive activity, suggesting an auto-inhibitory effect of the CRD on the receptor's activation[6,13]. Yet, the mechanism of how oxysterols activate Hh signalling and release CRD suppression remains unclear. Here, we present the crystal structure of the multi-domain human SMO in complex with a specially designed TMD ligand, shedding light on interactions between the CRD and TMD. Combining the structures with biophysical characterization and computer modelling results, we propose a mechanistic model of SMO activation.

## Results

**Rational design of a novel ligand for structural studies**. CRD truncated SMO (ΔCRD–SMO) has been previously co-crystallized with several small molecules, including both antagonists and agonists[3,8,9]. These ligands and others failed to yield crystals of the multi-domain SMO containing an intact CRD without stabilizing mutations. Analysis of previous ΔCRD–SMO structures[3,8,9] suggests that crystallization of a multi-domain SMO might require stabilization of some specific flexible parts of the structure. For example, we found that K395, located on top of the β-hairpin on extracellular loop 2, can engage in interactions with either adjacent residues or ligands[3], and thus probably adopt a dynamic conformational state (Fig. 1a). Thus, we hypothesized that further stabilization of this region by establishing a defined and stronger interaction might reduce the conformational heterogeneity.

For this purpose, we screened several commercial ligands using the CPM thermal shift assay[14] and chose LY2940680, which had the highest $T_m$ value, as our starting point for design of new ligands (Supplementary Fig. 1a). A series of compounds were then synthesized with modifications on the 2- and 4-positions of the benzoyl moiety of LY2940680 (Fig. 1b). Structure-activity relationship (SAR) analysis demonstrated the significance of these substitutions. First, removal of the 2-trifluoromethyl group lowered stability of the multi-domain SMO (TC101-104), indicating that the 2-trifluoromethyl group is essential and should be preserved (Fig. 1c). Next, several substituents were introduced by replacing the 4-fluorine atom to generate a stronger interaction between the ligands and adjacent residues. One of these ligands, TC114, with the 4-nitro group substitution, was predicted to form an electrostatic interaction with the protonated ε-amine of K395. Computer docking analysis

suggested that a distance of 3.4 Å between the amine and nitro groups would be right within the electrostatic or enhanced hydrogen-bond interaction distance. Consistent with the prediction, a CPM assay of LY2940680 analogues showed that TC114 confers a much stronger stabilizing effect on the multi-domain SMO ($T_m = 76$ °C) compared to LY2940680 ($T_m = 68$ °C) and other tested analogues (Fig. 1c; Supplementary Fig. 1a). Functional characterization of TC114 in a cell-based SMO activation assay confirmed the antagonist activity of TC114 on wild type (WT) SMO and reduced inhibition activity on the drug-resistant mutant (D473H)[15] (Supplementary Fig. 2a).

Analysis of our multi-domain SMO structures indicated that the nitro group introduced in the TC114 ligand does interact strongly with K395 in SMO as designed, and enhance the π–π stacking between the benzoyl group and F484 on helix VI (Fig. 1d). Therefore, we conclude that TC114 stabilizes the multi-domain SMO by keeping helix VI in a stable conformation that in turn enhances the hydrophobic interaction between the extended ECL3 and the CRD hydrophobic groove. Thus, the designed ligand TC114 achieved the desired goal, acting as a super stabilizing agent in the thermal stability assay and enabling the multi-domain SMO co-crystallization study.

**Overall architecture of the multi-domain human SMO**. To facilitate crystallization, human SMO was engineered by fusion of a Flavodoxin (FLA)[16] protein between helices V and VI, replacing part of intracellular loop 3 (ICL3) between P434 and S443. The N terminus 1–52 and C terminus 559–787 were truncated and a single-mutation E194M was made in the hinge domain (HD), which slightly increased the protein yield without affecting its stability compared to the construct without this mutation (Supplementary Figs 1b and 3). We also assessed the effect of this single point mutation on the receptor signalling activity, and noticed that E194M mutant showed some increased activity compared to the WT (Supplementary Fig. 2b). The resulting SMO–FLA construct was co-expressed with vismodegib[17], and then purified and crystallized in complex with TC114 using the lipidic cubic phase method[18]. We determined the structure using room temperature data collected at an X-ray free-electron laser (XFEL) source at 2.9 Å resolution (Supplementary Fig. 4). At the same time, we also solved the structure with data from 12 cryo-cooled crystals collected at a synchrotron source at 3.0 Å resolution (Supplementary Fig. 4). Since the room temperature XFEL crystal structure potentially represents a closer-to-native receptor conformation and at a slightly higher resolution than the synchrotron structure obtained at cryo-cooled conditions, we describe the XFEL SMO structure below, unless noted otherwise.

The overall SMO structure (Fig. 2a; Supplementary Fig. 4) shares a canonical GPCR 7TM bundle fold with an amphipathic helix VIII running parallel to the membrane plane. The CRD is positioned straight up on top of the TMD and is supported by extracellular loop 3 (ECL3) on one side and by a linker loop (residues 181–190) on the other. The CRD and TMD are connected by the HD (residues 191–220). Furthermore, on the extracellular side of the structure (Fig. 2b), helix VI is extended beyond the membrane surface by four α-helical turns, and the top part of helix VI is tilted at a non-proline kink towards the CRD, protruding into its hydrophobic groove and bridging a key connection between the CRD and TMD. A hydrophobic pocket is formed by the CRD (residues V107, L108, L112), HD (residue V210) and ECL3 (residues V494, I496) (Fig. 2c). This pocket was previously reported to constitute an oxysterol binding site[7,19].

**Unique multi-domain interaction**. As this manuscript was under preparation, Byrne *et al.* published two multi-domain SMO

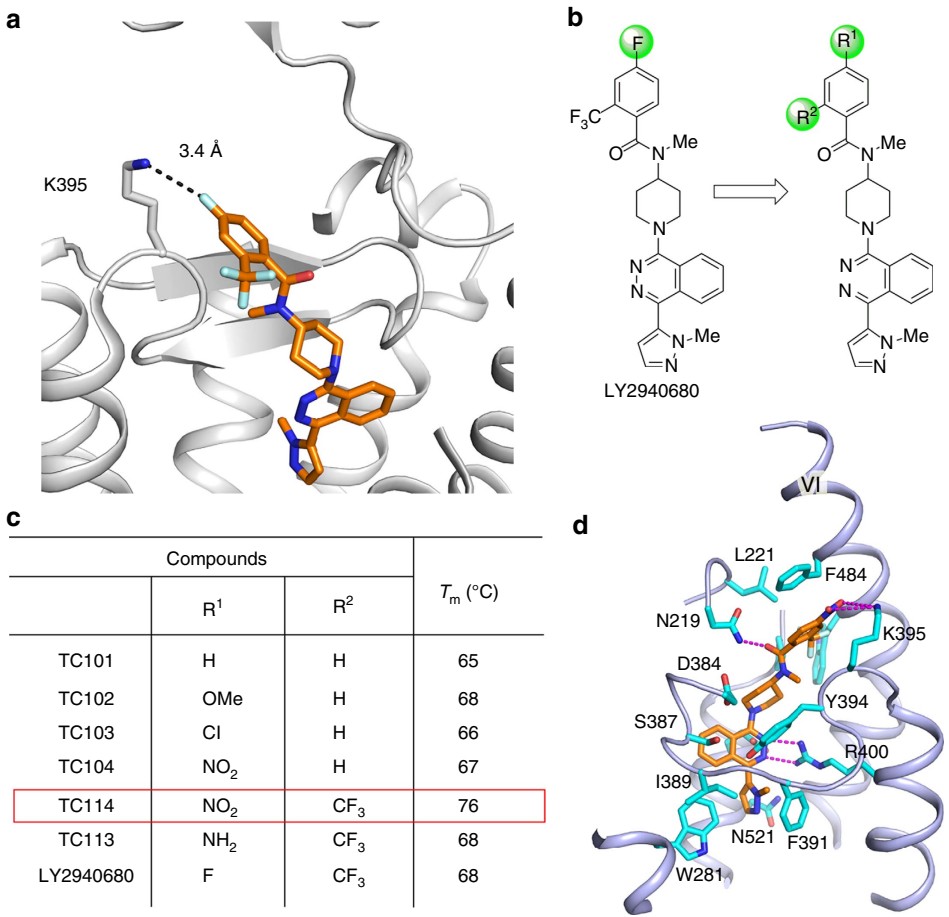

| Compounds | | | |
|---|---|---|---|
| | $R^1$ | $R^2$ | $T_m$ (°C) |
| TC101 | H | H | 65 |
| TC102 | OMe | H | 68 |
| TC103 | Cl | H | 66 |
| TC104 | $NO_2$ | H | 67 |
| TC114 | $NO_2$ | $CF_3$ | 76 |
| TC113 | $NH_2$ | $CF_3$ | 68 |
| LY2940680 | F | $CF_3$ | 68 |

**Figure 1 | Design and synthesis of SMO ligand TC114 for crystallographic studies.** (**a**) Close-up view of the LY2940680 binding pocket in ΔCRD–SMO structure (PBD ID: 4JKV). (**b**) Design and evolution of LY2940680 analogues for crystallization study by variation of the substituents on aromatic ring. (**c**) The representative $T_m$ values for LY2940680 and its analogues according to the CPM thermal shift assay. (**d**) Close-up view of TC114 binding pocket. TC114 (orange carbons) and SMO residues (cyan carbons) involved in ligand binding are shown in stick representation. The receptor is shown in light blue cartoon representation. Other elements are coloured as follows: oxygen, red; nitrogen, dark blue; sulfur, yellow. Hydrogen bonds are displayed as magenta dashed lines.

structures[20] and, therefore, we compared our XFEL structure with their vismodegib-bound and cholesterol-bound structures (Fig. 3a). Overall, the three structures are very similar with the TMD parts overlapping almost perfectly. The largest difference is in the mutual orientations of the CRD, ECL3 and HD. The CRDs in these three structures show substantial tilts between each other with respect to the aligned TMDs. In the vismodegib-bound structure, the CRD tilts by ∼3° towards the straight up conformation perpendicular to the membrane plane[20]. In the cholesterol-bound structure, however, the CRD tilts by ∼7° in the opposite direction along with an outward shift of both ECL3 and the upper part of helix VI compared to our XFEL structure (Fig. 3b). ECL3 in the cholesterol-bound structure makes an extra α-helical turn and shifts outward by as much as 7 Å compared to our XFEL structure (Fig. 3b), emphasizing the flexibility of this part of the structure and potential role of ECL3 in regulation of sterol binding to the CRD. Interestingly, in our XFEL structure, a glycan modification is observed on N493, which is also present in the vismodegib-bound and cholesterol-bound structures, although faced to a slightly different solvent side. In the vismodegib-bound structure, the glycan partially occupies the cholesterol binding site precluding cholesterol or oxysterol binding to the vismodegib-bound structure, whereas in our XFEL structure the side chain of I496 from ECL3 extends into the cholesterol binding site, also precluding cholesterol or oxysterol

binding to the CRD (Fig. 3b). This different glycan orientation is related to the aforementioned different ECL3 conformations between our XFEL structure and the cholesterol-bound structure. In addition, the HD in the cholesterol-bound structure moves towards the TM helices by ∼2 Å compared to the TC114-bound structure (Fig. 3c). Comparison of our XFEL multi-domain SMO structure with the five previously reported ΔCRD–SMO structures[3,8,9], and with the vismodegib-bound multi-domain SMO structure[20] showed that the TMD bundle and the TMD ligand binding pockets are highly consistent (Fig. 3d). The TMDs of these structures including loop regions are all very similar, except for different conformations of ECL3 and subtle differences at the intracellular end of helix V, likely related to different orientations of fusion partners in the crystal structures.

In the recently published work describing *Xenopus laevis* SMO (xSMO) CRD structures, the authors proposed that sterol binding induces a conformational change in the CRD (from 'open' to 'closed' conformation) and this conformational change is sufficient for SMO activation[21]. We compared SMO CRDs of the cholesterol-bound and our XFEL multi-domain SMO structures, with the three xSMO CRD structures in the apo state, bound to cyclopamine and bound to OHC (Fig. 3e). Binding of CRD agonists OHC or cyclopamine induces conformational changes in the CRD that mainly involve the displacement of key residues xW136 (hW163), xP137 (hP164), xF139 (hF166), xL140 (hL167)

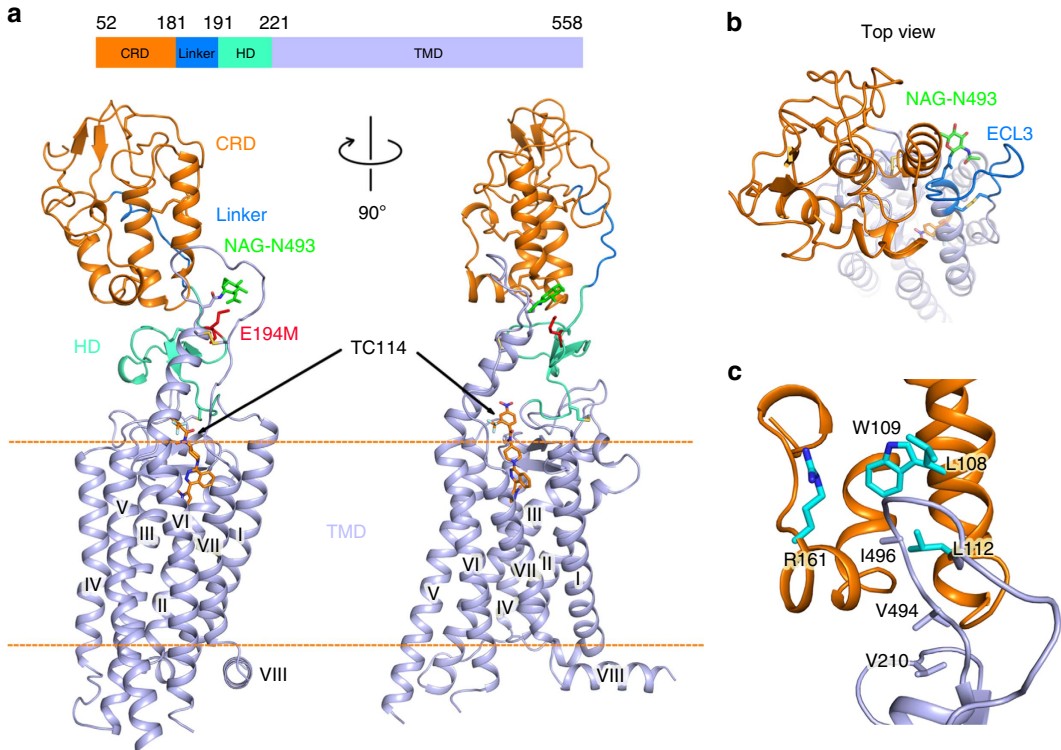

**Figure 2 | Overall structure of the multi-domain human SMO.** (**a**) Overall structure of the human SMO in complex with TC114 determined at an XFEL. TC114 is shown as orange sticks. The CRD, linker, HD and TMD are indicated as orange, marine, green cyan and light blue cartoons, respectively. The membrane boundary is labelled, as an orange dashed line. N-linked glycans (NAG) are shown in green sticks. (**b**) Top view of the SMO from the extracellular side. A hydrophobic pocket is formed by the CRD hydrophobic groove and ECL3 (marine loop). (**c**) Key residues in the CRD and ECL3 defining the hydrophobic pocket are shown in cyan and light blue sticks, respectively.

compared to the apo xSMO CRD structure[21]. Superposition of these CRD structures with CRDs of the multi-domain SMO structures showed that these key residues in both cholesterol-bound and our XFEL multi-domain SMO structures are in a conformation that is consistent with the OHC- or cyclopamine-bound, but not apo, CRD. This may indicate that conformational changes in the CRD are restricted, when it is placed in the context of the entire multi-domain SMO structure, where it always adopts a 'closed' conformation regardless of sterol binding. In fact, our structure demonstrates that in the absence of sterol binding, ECL3 interacts with the CRD hydrophobic groove to stabilize the CRD in a 'closed' conformation (Fig. 2c). The observation of limited, if any, conformational changes within the CRD itself on ligand binding is also supported by a recent publication by Luchetti *et al.*[12], where authors proposed a model of how cholesterol activates Hedgehog signalling through binding to SMO. Therefore, the conserved conformations of CRD itself, as observed in the multi-domain SMO structures may not be sufficient for SMO activation. Conformational modularity on the CRD anchoring region—ECL3, instead, is rather critical. Point mutations on ECL3, including N493Q and I496R, lead to SMO constitutive activity, consistent with the self-inhibitory role of CRD to the receptor basal activity[11] (Supplementary Fig. 2b).

To probe the flexibility of SMO 'hotspots' in solution, we performed hydrogen-deuterium exchange (HDX) analysis using the purified SMO protein with or without ligands. HDX occurs in solvent accessible parts of the protein, so changes in conformation on ligand binding that expose or mask protein regions can be measured by the altered hydrogen-deuterium exchange rates. Interestingly, we did not observe any changes in the solvent deuterium uptake kinetics, when comparing the apo

receptor to either the CRD agonist OHC-bound or CRD antagonist 22-azacholesterol (NHC)-bound receptors. Adding TMD ligand TC114, however, induced protection to exchange (slower exchange) at regions mostly on the cytoplasmic side of the receptor (Supplementary Fig. 5). This analysis demonstrates that although no TMD rearrangements were observed on the cytoplasmic side in any SMO crystal structures solved to date, the possibility that this region is intrinsically dynamic and subject to conformational changes on signal transduction could not be ruled out. Similarly, in a 1 μs-long molecular dynamics (MD) simulations with SMO in the presence of cholesterol and embedded in a lipid environment (Supplementary Fig. 6), we do not see large degree of movement on the CRD nor TMD domains, suggesting that these domains are rather constrained when being activated and the structural stability is important for its functionality. This MD result is interesting, as we originally expected higher flexibility on the CRD region; yet it is consistent with our observation from the HDX analysis, as well as with the 100-ns MD result reported by Byrne *et al.*[20] Our longer-time MD simulation showed that cholesterol-bound SMO CRD leans towards the membrane plane, in agreement with the observation from our structural comparison between TC114-bound and cholesterol-bound SMO structures (Fig. 3a). Further investigation along this line is required to draw more comprehensive conclusion, particularly regarding the degree of freedom on the multi-domains.

**Comparison of human SMO with other class frizzled receptors.** Superposition of the CRDs between human SMO and mouse Frizzled-8 (mFzd-8) or human Frizzled-4 (hFzd-4) receptors

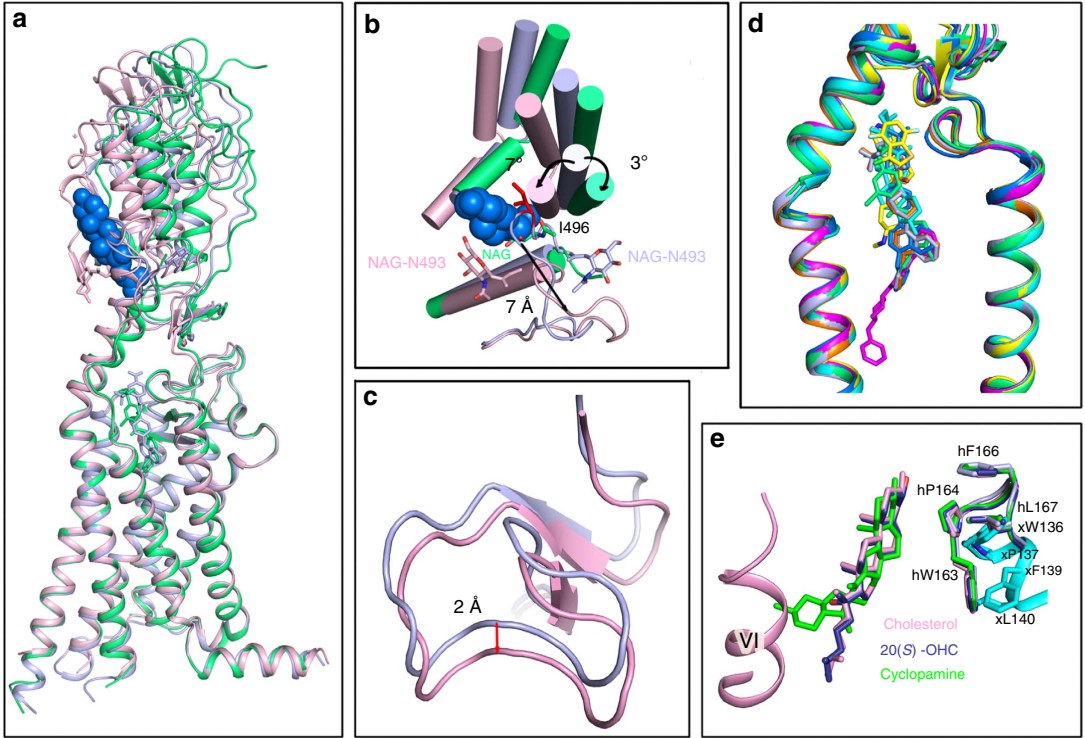

**Figure 3 | Unique multi-domain interaction and modularity in the human SMO.** (**a**) Superposition of three multi-domain SMO structures: SMO solved at XFEL, SMO in complex with cholesterol (PBD ID: 5L7D) and SMO in complex with vismodegib (PBD ID: 5L7I) are shown in light blue, pink and lime green cartoons, respectively. The cholesterol in the 5L7D structure is shown in marine spheres. (**b**) Top view of the extracellular side. The glycans from XFEL, cholesterol-bound and vismodegib-bound structures are shown in light blue, pink and lime green sticks, respectively. I496 from XFEL structure is shown in red sticks. (**c**) Hinge domains (HDs) from SMO in complex with TC114 (light blue) and SMO in complex with cholesterol (PBD ID: 5L7D; pink) are superimposed and shown as cartoons. (**d**) Close-up view of superimposed structures of the TMD bundle of SMO with different ligands bound. SMO in complex with TC114 (light blue), SMO in complex with vismodegib (PBD ID: 5L7I; lime green), ΔCRD–SMO in complex with LY2940680 (PBD ID: 4JKV; orange), ΔCRD–SMO in complex with Anta XV (PDB ID: 4QIM; marine), ΔCRD–SMO in complex with SAG 1.5 (PDB ID: 4QIN; yellow), ΔCRD–SMO in complex with SANT1 (PDB ID: 4N4W; magenta) and ΔCRD–SMO in complex with Cyclopamine (PDB ID: 4O9R; cyan) are superimposed and shown as cartoons. (**e**) Superimposed structures of SMO CRDs. CRD from multi-domain SMO structure solved at XFEL (light blue), CRD from multi-domain SMO structure in complex with cholesterol (PBD ID: 5L7D; pink), CRD from xSMO in the apo state (PBD ID: 5KZZ; cyan), CRD from xSMO in complex with 20(S)-OHC (PBD ID: 5KZV; deep blue) and CRD from xSMO in complex with cyclopamine (PBD ID: 5KZY; green) are shown in cylindrical helices. Key residues, cholesterol, 20(S)-OHC and cyclopamine are shown in sticks.

(Fig. 4a) reveals a highly conserved architecture and highlights a closely related role of the CRD for ligand recognition. Moreover, sequence alignment of all human Class Frizzled GPCRs (SMO and ten Frizzled receptors) indicates the existence of a hydrophobic groove (site 1) in all CRDs, and a hydrophobic patch (site 2) on the opposite side of the CRD[22,23], which exists only in Frizzled receptors (Supplementary Fig. 7). In the crystal structure of mFzd-8 CRD in complex with its native ligand homologue xWnt8, a palmitoyl group modification on xWnt8 was discovered bound to the hydrophobic groove (site 1) (Fig. 4b). The key residues, Q71, F72, P74, L75, I78, M122, Y125, F127 of mFzd-8, which form this site 1 hydrophobic groove, are highly conserved within all Class Frizzled receptors including SMO. At the hydrophobic patch site 2, Norrin, a secreted retinal vascular growth factor, recognized as an endogenous ligand only for Fzd-4, uses an extended hydrophobic interface to interact with Fzd-4's CRD[23] (Fig. 4c). The hydrophobic patch is formed by residues from hFzd-4 CRD F96, M105, I110, M157 and M159. In the mFzd-8 CRD–xWnt8 structure, a long finger loop from Wnt also extends into the same hydrophobic patch interacting with site 2. This hydrophobic patch (or site 2) is conserved in all 10 Frizzled receptors. Surface hydrophobicity analysis indicates that while the interface of hFzd-4 and Norrin is very hydrophobic, the same site is hydrophilic in SMO (Fig. 4c). The absence of a

hydrophobic patch in SMO, however, argues against the existence of a hydrophobic binding mode in this site, and indicates that site 1 may be the only binding site for endogenous sterols or yet unidentified sterol-modified protein ligands.

## Discussion

The mechanism of how the binding of an agonist to the CRD triggers a conformational change in the TMD remains obscure. As noted, binding of cholesterol to the multi-domain SMO induces an ~7° tilt of the CRD, which is associated with three major conformational changes: the extracellular extension of helix VI tilts outward by ~5°, ECL3 forms an extra α-helical turn and shifts outward by 7 Å, and the HD is displaced towards the TMD by 2 Å compared to the TC114-bound structure (Fig. 3a–c). In the xSMO cyclopamine-bound CRD structure, the E and F rings of cyclopamine, which acts as an agonist when bound to the CRD, clash with helix VI of the cholesterol-bound multi-domain SMO structure, indicating that helix VI and ECL3 should shift outward even further when cyclopamine is bound to the CRD in a multi-domain SMO (Fig. 3e). Structure-guided mutagenesis on HD, V198R and K204A, which disrupted the conformational integrity of HD, can inhibit 20(S)-OHC induced signalling to these mutants without affecting Sonic Hedgehog (Shh) induced

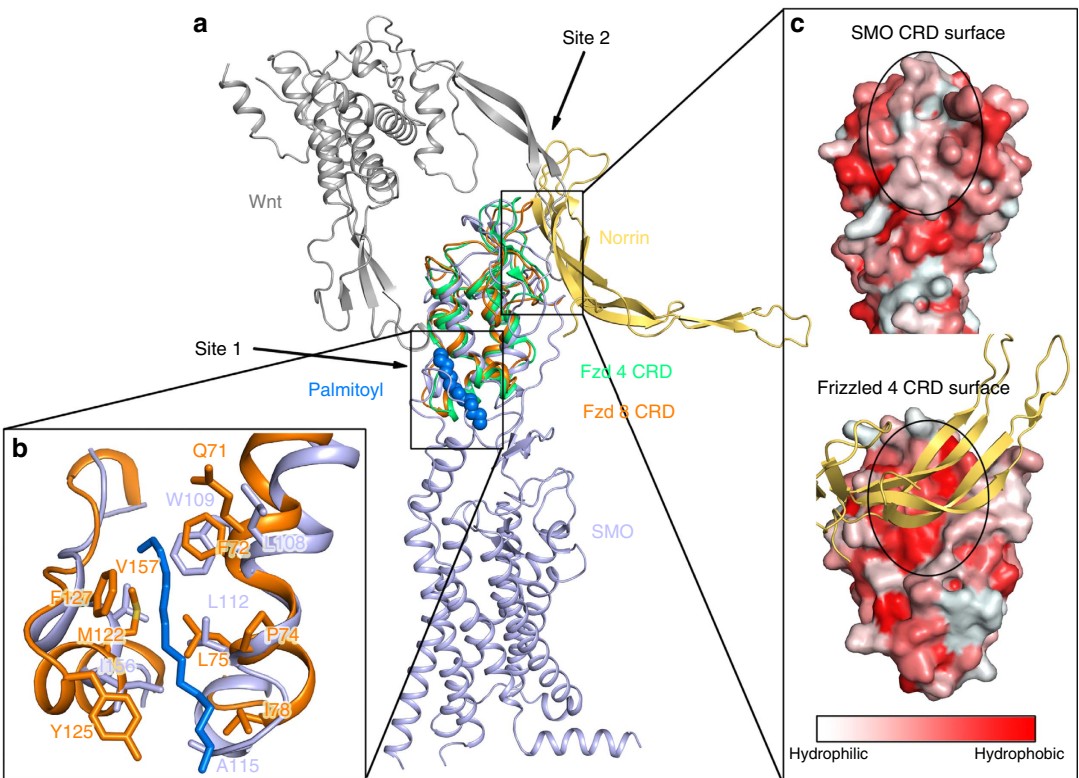

**Figure 4 | Comparison of the CRD of the human SMO with the Frizzled receptors.** (**a**) Side view of superimposed structures of the human SMO (hSMO) CRD with the hFzd-4 (PDB ID: 5CL1) and mFzd-8 CRDs (PDB ID: 4F0A). The SMO, hFzd-4 and mFzd-8 are shown as cartoons in light blue, lime green and orange, respectively. The Wnt and Norrin are shown as dark grey and yellow orange cartoons, respectively. The palmitoyl group in the mFzd-8 CRD is shown in marine spheres. (**b**) Site 1: Close-up view of the palmitoyl group with interacting residues as orange sticks. The palmitoyl group is shown in marine sticks. The residues from SMO forming the hydrophobic pocket are shown in light blue sticks, mFzd-8 in orange sticks. (**c**) Site 2: Surface of SMO and hFzd-4 CRD. Norrin is shown in light orange cartoons. The colour gradient from light red to dark red corresponds to the change of surface property from hydrophilic to hydrophobic. The Norrin binding site on hFzd-4 CRD surface is labelled by a black dashed circle with corresponding site also marked on the SMO surface.

signalling, suggesting a modulatory role of this region to SMO activation by oxysterols (Supplementary Fig. 8). This result is also consistent with a previous report that 20(*S*)-OHC activates SMO in a distinct way compared to endogenous signalling[12]. Meanwhile, our HDX analysis indicated the existence of an intrinsic flexibility at the cytoplasmic side of the TMD, which could allow for further conformational changes on signal transduction. We suggest that the outward tilt of upper helix VI and ECL3, and the displacement of the HD are likely to be responsible for triggering a conformational change in the receptor and activating it. However, this hypothesis still requires further verification.

It has been shown that the CRD exerts allosteric modulation of the TMD[11], and vice-versa, affecting ligand recognition and receptor activation. Similarly, the glucagon receptor (GCGR), a Class B GPCR, also employs a multi-domain architecture for ligand recognition and activation. GCGR has unambiguously distinct states defined as an 'open/closed' switch, as revealed by computer modelling of the full length receptor[24]. Extensive biophysical and computational analysis of GCGR have proposed an activation mechanism for GCGR on glucagon binding. In the absence of the endogenous peptide ligand glucagon, GCGR adopts a dynamic conformation, where the ECD, an equivalent counterpart of the CRD in GCGR, undergoes a large-scale swinging motion between an open state, in which the ECD points into the extracellular space away from the membrane, and a closed state, in which the ECD lays on top of the ECLs of the TMD. When glucagon binds to GCGR, it stabilizes the receptor in

the 'open' conformational state, where the ECD is oriented almost perpendicular to the membrane surface allowing the N-terminus of glucagon to enter in the TMD binding site and activate the receptor. In SMO, by contrast, the range of CRD motion appears to be limited in the consideration of the structure analysis and HDX results. Agonist binding to the CRD first triggers a small tilt of the CRD, which pushes helix VI and ECL3 outward. Such a combined tilt possibly leads to an amplified conformational change along with a downward movement of the HD, thereby transmitting the signal to the TMD and activating the receptor. However, no such TMD conformational changes have yet been observed in any SMO structure, possibly due to the ICL3 fusions employed or crystal packing. Thus, capturing a fully active SMO conformation may require the presence of an intracellular binding partner to which signal is transmitted.

SMO is a validated target for anti-cancer drugs. There are two FDA approved drugs acting through regulation of SMO mediated signalling pathway: vismodegib (GDC-0449) (ref. 17) and sonidegib (LDE-225) (ref. 25) for the treatment of basal cell carcinoma. Both are SMO antagonists bound to TMD. Long-term administration of these drugs, however, can lead to the development of resistance[26,27]. In this work, we applied a successful strategy towards the design of a new tool compound for the crystallographic study of a challenging GPCR. As a drug candidate in phase 2 clinical trials with Eli Lilly, LY2940680 is being employed to treat small-cell lung cancer[28], and was the first ligand used successfully in a SMO structural study. Nevertheless, for crystallization of the multi-domain SMO, LY2940680 had to

**Table 1 | Crystallographic data collection and refinement statistics.**

**Data collection and refinement statistics**

| | fISMO_XFEL | | fISMO_Synchrotron | |
|---|---|---|---|---|
| Data set | fISMO_XFEL | | fISMO_Synchrotron | |
| Temperature (°C) | 20 (Room temperature) | | − 196 | |
| Space group | P2$_1$ | | P2$_1$ | |
| | | | | |
| *Cell dimensions* | | | | |
| a,b,c (Å) | 40.6 349.5 61.8 | | 40.1 356.4 59.1 | |
| β (deg) | 101.1 | | 102.8 | |
| Number of reflections measured | 13,583,207 | | 109,498 | |
| Number of unique reflections | 37,101 | | 29,571 | |
| Resolution (Å) | 24.90–2.90 (3.00–2.90) | | 48.38–3.00 (3.16–3.00) | |
| $R_{merge}$ or $R_{split}$ | 13.3 (280) | | 11.7 (39.3) | |
| Mean $I/\sigma$ (I) | 5.4 (0.4) | | 7.5 (1.9) | |
| Completeness (%) | 100 (100) | | 91.9 (81.3) | |
| Redundancy | 366 (64.5) | | 3.7 (2.3) | |
| CC* | 0.9986 (0.584) | | 0.998 (0.875) | |
| | | | | |
| *Refinement* | | | | |
| Resolution (Å) | 24.90–2.90 | | 48.38–3.00 | |
| Number of reflections (test set) | 37,045 (1,810) | | 29,552 (1,502) | |
| $R_{work}/R_{free}$ | 0.218/0.239 | | 0.203/0.240 | |
| | | | | |
| *Number of atoms* | | | | |
| | A | B | A | B |
| SMO | 3,770 | 3,778 | 3,878 | 3,871 |
| Flavodoxin | 1,099 | 1,100 | 1,103 | 1,097 |
| TC114 | 39 | 39 | 39 | 39 |
| Other | 45 | 31 | 31 | 43 |
| | | | | |
| *Average B Factor (Å$^2$)* | | | | |
| Wilson/Overall | 82.2/117.3 | | 72.6/82.7 | |
| | A | B | A | B |
| SMO | 127.1 | 121.8 | 93.2 | 87.3 |
| Flavodoxin | 97.8 | 90.8 | 59.6 | 55.0 |
| TC114 | 93.6 | 105.5 | 82.5 | 74.4 |
| Other | 106.2 | 63.3 | 52.8 | 55.1 |
| | | | | |
| *r.m.s.d.'s* | | | | |
| Bond lengths (Å) | 0.009 | | 0.009 | |
| Bond angles (deg) | 1.03 | | 0.93 | |
| | | | | |
| *Ramachandran plot statistics (%)** | | | | |
| Favored regions | 95.2 | | 94.1 | |
| Allowed regions | 4.8 | | 5.9 | |
| Disallowed regions | 0 | | 0 | |

Data for high resolution shells is shown in parenthesis where applicable.
*As defined in MolProbity[52].

be modified to stabilize the complex CRD–TMD interaction. Four-fluorine was originally introduced in the benzoyl group of LY2940680 to potentially block the rapid metabolism that is a usual strategy to optimize ADME property in drug discovery. This substitution was however detrimental to the stability and crystallization of the receptor. Although medicinal chemists generally try to avoid the nitro group, especially attached to an aromatic ring, which is known to be easily reduced as well as to carry toxic effects, the design of the TC114 tool compound by the replacement of the 4-fluorine with the 4-nitro group in LY2940680 has been one of the key factors that led to the multi-domain SMO structure determination. We therefore are optimistic about the possibility of extending this design strategy to other in-progress receptors. The multi-domain SMO structure may inspire the design of a new type of small molecule that links the CRD and TMD, or interacts with the HD to regulate domain-domain communications that can potentially exert amplified efficacy and overcome drug resistance. Alternatively, drugs designed for CRD binding grooves could also be explored to circumvent the resistance resulting from mutations in the TMD. The structure offers insights into the conformational modularity of SMO multi-domains and provides new opportunities for the design of ligands to battle SMO-related diseases.

## Methods

**Synthesis of TC114.** A solution of 4-nitro-2-(trifluoromethyl)benzoic acid (80 mg, 0.34 mmol), secondary amine *N*-methyl-1-(4-(1-methyl-1*H*-pyrazol-5-yl) phthalazin-1-yl)piperidin-4-amine (100 mg, 0.31 mmol) and *N*,*N*-diisopropylethy-lamine (DIPEA, 60 mg, 80 μl, 0.46 mmol) in 3 ml of $CH_2Cl_2$ were treated with 1-[bis(dimethylamino)methylene]-1H-1,2,3-triazolo[4,5-b]pyridinium 3-oxid hexafluorophosphate (HATU, 153 mg, 0.40 mmol). The reaction mixture was stirred at room temperature for 1 h before being quenched by the addition of brine. The reaction mixture was extracted three times with $CH_2Cl_2$. The combined organic layer was washed with saturated $NaHCO_3$ solution and brine sequentially, then dried over $Na_2SO_4$. After filtration, the solution was concentrated in vacuum and the crude product was purified by flash column chromatography on silica gel to yield TC114, as a colourless solid (82 mg, 49%). Nuclear magnetic resonance showed a mixture of rotamers.

**Engineered SMO-FLA fusion construct for structural studies.** The wild-type (WT) human SMO gene was synthesized by Genescript and then cloned into a modified pTT5 vector containing an expression cassette with an HA signal sequence followed by a FLAG tag, a $10 \times$ His tag, and a tobacco etch virus (TEV) protease recognition site at the N terminus before the receptor sequence and another $10 \times$ His tag at the C-terminus. A small protein Flavodoxin (FLA, MW 16KD)[16] was fused to ICL3 between P434 and S443, using overlapping PCR. The N terminus 1–52 and C terminus 559–787 were truncated. A single-mutation E194M in the HD was introduced for the construct, crystals of which were used for data collection at XFEL. The construct used for synchrotron structure determination does not contain the point mutation and was slightly modified by further truncating the N terminus by five residues. All primer sequences used in this study are shown in Supplementary Table 1.

**Expression and purification of SMO-FLA fusion protein.** The engineered SMO construct was expressed in HEK293F cells (Invitrogen) in the presence of $5\,\mu M$ vismodegib. HEK293F cells at a cell density of $1.0 - 1.3 \times 10^6$ cells $ml^{-1}$ were transiently transfected with PEI:DNA at a ratio of 2:1, and cultured at $37\,°C$. Cells were collected 48 h, after transfection and stored at $-80\,°C$ until use. Cell pellets were re-suspended in a hypotonic buffer (10 mM HEPES, pH 7.5, 10 mM $MgCl_2$, 20 mM KCl and EDTA-free protease inhibitor cocktail tablets (Roche)). Further washing of the raw membranes was performed by repeated centrifugation (three times) in a high salt buffer (10 mM HEPES, pH 7.5, 10 mM $MgCl_2$, 20 mM KCl, 1.0 M NaCl and EDTA-free protease inhibitor cocktail tablets). The washed membranes were re-suspended in a buffer containing 30 μM TC114, 2 mg $ml^{-1}$ iodoacetamide (Sigma) and EDTA-free protease inhibitor cocktail tablets, and incubated on a rocker at $4\,°C$ for 1 h. The membranes were then solubilized in a buffer containing 50 mM HEPES, pH 7.5, 200 mM NaCl, 1% (w/v) n-dodecyl-β-D-maltopyranoside (DDM, Anatrace) and 0.2% (w/v) cholesteryl hemisuccinate (CHS, Sigma), for 2.5 h at $4\,°C$. After high-speed centrifugation, the supernatant was incubated with TALON IMAC resin (Clontech) overnight at $4\,°C$ supplemented with 20 mM imidazole and 1.0 M NaCl. After binding, the resin was washed with 10-column volumes of wash I buffer (50 mM HEPES pH 7.5, 800 mM NaCl, 10% glycerol, 0.5% (w/v) Lauryl Maltose Neopentyl Glycol (LMNG, Anatrace)/0.1% CHS, 20 mM Imidazole, 10 mM $MgCl_2$, 6 mM ATP and 30 μM TC114). The beads with 2 ml wash I buffer were transferred to a 5 ml tube and incubated on a rocker at $4\,°C$ for 2 h for complete detergent exchange, followed by washing with six-column volumes of wash II buffer (25 mM HEPES pH 7.5, 500 mM NaCl, 10% glycerol, 0.03% LMNG/0.006% CHS, 40 mM Imidazole and 50 μM TC114). The protein was then eluted by three-column volumes of elution buffer (25 mM HEPES pH 7.5, 300 mM NaCl, 10% glycerol, 0.01% LMNG/0.002% CHS, 220 mM Imidazole and 100 μM TC114). The protein was then treated overnight with TEV protease and Endo H (NEB) to cleave the N-terminal His tag, FLAG tag and additional glycans. The protein was then concentrated to 40-50 mg $ml^{-1}$ with a 100 kDa cutoff Vivaspin concentrator. Protein monodispersity was tested by analytical size-exclusion chromatography (aSEC). Typically, the aSEC profile showed a monodisperse peak.

**Crystallization in LCP for synchrotron data collection.** Protein samples of the SMO receptor in a complex with TC114 were reconstituted into lipidic cubic phase (LCP) by mixing with molten lipid (10% (w/w) cholesterol, 90% (w/w) monoolein) at a ratio of 2/3 (v/v) protein solution/lipid using a mechanical syringe mixer[29]. LCP crystallization trials were performed using an NT8-LCP crystallization robot (Formulatrix)[30] in 96-well glass sandwich plates (Nova). After setup, plates were incubated and imaged at $20\,°C$ using an automated incubator/imager (RockImager 1000, Formulatrix). Initial crystal hits were found in a precipitant condition containing 100 mM Sodium citrate tribasic dihydrate, pH 5.0, 30% (v/v) PEG400, 100 mM Ammonium nitrate. After optimization, crystals grew in 100 mM Sodium citrate tribasic dihydrate pH 5.0, 36% (v/v) PEG400, 50–200 mM Ammonium nitrate to the average size of $47 \times 23 \times 8\,\mu m^3$ within 7 d. The SMO crystals were collected directly from LCP using 30 μm micromounts (MiTeGen) and flash frozen in liquid nitrogen for data collection.

**Synchrotron data collection and structure determination.** X-ray diffraction data were collected at the SPring-8 beam line 41XU, Hyogo, Japan, using a Pilatus3 6M detector (X-ray wavelength 10,000 Å). The crystals were exposed with a 10 μm mini-beam for 0.2 s and 0.2° oscillation per frame. XDS[31] was used for integrating, scaling and merging data from 12 best-diffracting crystals of the SMO–TC114 complex. Initial phase information of the SMO-TC114 complex was obtained by molecular replacement (MR) with Phaser[32] using the TMD of human–SMO (PDB ID: 4QIM)[8], CRD of zebrafish SMO (PDB ID: 4C79)[33], and a Flavodoxin structure (PDB ID: 1I1O)[34] as search models. The correct MR solution contained two SMO molecules packed tail-to-tail in one asymmetric unit of the $P2_1$ lattice. Refinement was performed with REFMAC5 (ref. 35) and autoBUSTER[36] followed by manual examination and rebuilding of the refined coordinates in the program COOT[37] using both |2Fo|−|Fc| and |Fo|−|Fc| maps. NCS and TLS refinement with two TLS groups (SMO and Flavodoxin domains) were incorporated in the refinement. The final model of the synchrotron SMO-TC114 complex contains 498 residues of SMO (residues 58–433 and 444–565) and the 147 residues of

Flavodoxin in molecule A, and 494 residues of SMO (residues 58–433 and 444–561) and 147 residues of Flavodoxin in molecule B. Data collection and refinement statistics of the synchrotron structure are summarized in Table 1.

**Crystallization in LCP for XFEL data collection.** Crystals for LCP–XFEL were obtained in Hamilton gas-tight syringes as previously described[38] by injecting $\sim 5\,\mu l$ of protein-laden LCP as a continuous column of $\sim 400\,\mu m$ in diameter into a 100 μl syringe filled with 60 μl of precipitant solution containing 100 mM Sodium citrate tribasic dihydrate, pH 5.0, 36% (v/v) PEG400, 150 mM Ammonium Chloride and incubated for at least 24 h at $20\,°C$. After crystals (average size $5 \times 5 \times 2\,\mu m^3$) formed, samples from 2 to 3 syringes were consolidated together and the excess of precipitant solution was removed. The residual precipitant solution was absorbed by addition of a few microliter of molten 7.9 MAG lipid[39]. The resulting crystal-laden LCP sample was inspected under a visual microscope and loaded in an LCP injector[9] for LCP-SFX data collection.

**XFEL data collection and structure determination.** The experiment was carried out at the Coherent X-ray Imaging end station[40] of the LCLS, using 9.5 keV (1.3 Å) X-ray pulses with a pulse duration of 45 fs and a repetition rate of 120 Hz. SMO–TC114 crystals were delivered to the intersection with a 1.5 μm XFEL beam inside a vacuum chamber within a 50 μm diameter stream of LCP at a flow rate of $\sim 200$ nl $min^{-1}$ generated by an LCP injector. Serial femtosecond crystallography (SFX) data were collected by a CSPAD detector positioned at a distance 106 mm from the sample. The beam was attenuated to 7.5–11% ($\sim 10^{11}$ photons per pulse) of the full intensity to avoid detector saturation. A total of 2,102,907 diffraction patterns were collected, of which 320,121 were identified as potential single crystal diffraction hits with $>15$ potential Bragg peaks by the software Cheetah[41], corresponding to an average hit rate of 15.2%. Auto indexing and structure factor integration of the crystal hits was performed using CrystFEL (version 0.6.2)[42] with a 'pushres 1.2' option, resulting in 65,560 indexed images with a monoclinic lattice (20.5% indexing success rate). Initial phase information for the XFEL structure of the SMO–TC114 complex was obtained by molecular replacement (MR) with Phaser[32] using our synchrotron structure with CRD, TMD and Flavodoxin domains as independent search models. An MR trial with the multi-domain SMO synchrotron structure as a single search model was not successful. The correct MR solution contained two SMO–TC114 molecules packed tail-to-tail in one asymmetric unit of the $P2_1$ lattice, similar to the synchrotron structure. Refinement was performed with REFMAC5 (ref. 35) and autoBUSTER[36] followed by manual examination and rebuilding of the refined coordinates in the program COOT[37] using both $|2F_o| - |F_c|$ and $|F_o| - |F_c|$ maps. NCS restraints were used during the refinement. Addition of TLS groups decreased the quality of the $R/R_{free}$ statistics and, therefore, TLS was not used in the refinement. The final model of the SMO–TC114 complex contains 485 residues of SMO (residues 59–433 and 444–553) and the 147 residues of Flavodoxin (residues 1002–1148) in molecule A, 487 residues of SMO (residues 59–433, 444–498 and 505–561) including uncleaved C-term Histag (559–561) and the 147 residues of Flavodoxin (residues 1002–1148) in molecule B. The remaining N-terminal residues (residues 53–58 in molecules A and B), part of ECL3 (residues 499–504 in molecule B) and C-terminal residues (554–558 in molecule A) are likely disordered and not visible in the electron density maps, and therefore are not modelled. Data collection and refinement statistics of the XFEL structure are summarized in Table 1.

**Hydrogen-Deuterium exchange mass spectrometry analysis.** Hydrogen-Deuterium exchange mass spectrometry (HDX-MS) employed a SMO construct similar to the crystallization construct (SMO-FLA), except that the fusion partner was replaced by a smaller protein Rubredoxin (RUB, MW 6KD) inserted into ICL3 between P434 and S443 (SMO-RUB), and the experiment was carried out at $4\,°C$ as follows. Briefly, 25 μM of purified protein as apo (no ligand) was mixed with 250 μM of TC114, or with 250 μM of OHC, or with 250 μM of NHC and incubated in a $D_2O$ buffer for a range of exchange times from 10 s to 1 h before quenching the deuterium exchange reaction with an acidic quench solution (pH 2.4). All mixing and digestions were carried out on a LEAP Technologies Twin HTS PAL liquid-handling robot housed inside a temperature-controlled cabinet[24,43]. Digestion was performed in line with chromatography using an immobilized pepsin column. Mass spectra were acquired on a Q Exactive hybrid quadrupole-Orbitrap mass spectrometer (Thermo Scientific) and per cent deuterium exchange values for peptide isotopic envelopes at each time point were calculated and processed using the HDX Workbench software[44]. The data are presented as an average ± s.d. of three independent replicates. Data were fitted to a simple nonlinear regression (least squares) best fit model (X is log and Y is linear) using GraphPad Prism.

**MD simulation.** *Protein preparation and system building.* The crystal structure of SMO receptor with cholesterol was obtained from the PDB database (PDB code: 5L7D)[20]. The chain A of the structure was selected to build the model. Prime[45] in Schrödinger Release 2015-3 was used and ICL3 was built by a homology model using another crystal structure of Smo (PDB code: 4JKV (ref. 3)), and ICL2 loop was predicted by Prime. The model was refined by Prepwizard[46] in pH at 7.0, and the C- or N-terminal was capped by ACT and NME.

The membrane around the transmembrane domain of SMO receptor was built by 135 POPC molecules and 48 cholesterols using CHARMM-GUI web server[47]. The final periodic boundary system box is 84.75 × 84.75 × 139.85 Å, and 20881 TIP3P waters and 62 Na$^+$ and 68 Cl$^-$ ions (0.15 mol l$^{-1}$ NaCl) were used to solvate and neutralize the box. The system had a total of 93,475 atoms per periodic cell.

*Molecular dynamic simulation and data analysis.* The Amber99sb-ILDN[48] force filed was applied on SMO receptor, TIP3P water and ions, and the Slipids[49] force field was applied for cholesterol and POPC. Gromacs 5.1.2 (ref. 50) was used for the simulation, and the relaxation protocol was obtained from CHARMM-GUI and described in their paper and web server. First, the system is minimized in 10,000 steps by steepest-descent procedure, and equilibrated at constant temperature (303.15 K) and constant pressure (1.0 bar) in 50 ps NVT condition and 125 ps NPT condition with strong restraints on protein and lipids and a nonbonded cutoff switching ranges of 10–12 Å. Subsequently, 2 ns NPT equilibration in the same condition with weak restraints of protein and lipids was performed to get the equilibrated phase. Finally, two identical 1,000 ns product MD simulations were performed on the SMO receptor and cholesterol as ligand in lipids and water system.

The result of product MD simulation was analysed by Gromacs 5.1.2. The angle defined by three atoms (the C-alpha of P69, V210 and W535) was selected to describe the tilt of SMO CRD relative to SMO TMD.

**Hedgehog signalling assay.** Light II cells (ATCC; Rockville, MD) with stable ectopic expression of 8 × Gli binding site-firefly luciferase and constitutive Renilla luciferase reporter constructs were seeded into 96-well plates. After transfection with either Smoothened wild type or distinct Smoothened mutant plasmids as indicated, the cells are subjected to various treatments as indicated for 36 h. Luciferase activity in light II cells was measured using the Dual-Luciferase Assay System kit from Promega (according to the manufacturer's instructions in a luminometer (Molecular Devices; Sunnyvale, CA)) (Madison, WI), and was normalized to Renilla values. Data were plotted, and IC50 values were determined using GraphPad Prism. Each data point represents the mean ± s.d. repeated in triplicates.

**Western blot analysis.** Cells after various transfections as indicated were collected and subjected to lysis buffer (50 mM Tris, pH 7.4, 150 mM NaCl, 1% NP-40, 1 mM sodium vanadate, 1 mM PMSF, 1 mM DTT, 10 mg ml$^{-1}$ of leupeptin and aprotinin), followed by immunoblot analysis. Primary antibodies against SMO and GAPDH (Santa Cruz Biotechnology, Santa Cruz, CA, USA) were used for immunoblot analysis according to the routine procedure[51]. The antibodies against SMO and GAPDH are diluted at 1:200 and 1:5,000, respectively, according to manufacturer's instruction.

**Data availability.** Coordinates and structure factors have been deposited in the Protein Data Bank for SMO-TC114 solved at XFEL (PDB: 5V56) and at synchrotron (PDB: 5V57). The PDB accession codes 4JKV, 5L7D, 5L7I, 4QIM, 4QIN, 4N4W, 4O9R, 5KZZ, 5KZV, 5KZY, 5CL1, 4F0A were used in this study. The UniProt accession codes Q99835 for human SMO was used in this study. All other data are available from the corresponding authors on reasonable request.

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

## Acknowledgements

This work was supported by the 'Pujiang Talents' grant 14PJ1406700 from the Science and Technology Commission of Shanghai Municipality and the National '1000 Talents' young scientist grant (F.X.), and in part by the National Institutes of Health (NIH) grant R01 GM108635 (V.C.) and the National Science Foundation (NSF) grant 1231306 (V.C.). C.G. kindly thanks the Human Frontier Science Program (LT000087/2015-L) for financial support. Parts of the crystal delivery system used at LCLS for this research was funded by the NIH grant P41GM103393. We thank the Shanghai Municipal Government, ShanghaiTech University for financial support (F.X., H.T., S.Z. and R.C.S.). Use of the Linac Coherent Light Source (LCLS), SLAC National Accelerator Laboratory, is supported by the US Department of Energy, Office of Science, Office of Basic Energy Sciences under Contract No. DE-AC02-76SF00515. The synchrotron radiation experiments were performed at the BL41XU of SPring-8 with approval of the Japan Synchrotron Radiation Research Institute (JASRI) (proposal no. 2015B1025 and 2016A2725). We thank Angela Walker for assistance with manuscript preparation; Martin Audet for assistance with large-scale protein purification for HDX experiments; Tao Li for assistance with initial synthesis of key intermediates; Xiaoyan Liu for assistance with cell-based functional assay; and Suwen Hu, Xiaowen Li, Qiaoyun Shi at the Mammalian Expression Core Facility of iHuman Institute for their support.

## Author contributions

X.Z. optimized the construct, developed the purification procedure and purified the SMO proteins for crystallization, performed crystallization trials and optimized crystallization conditions, collected synchrotron diffraction data, analysed the data, prepared the figures and wrote the manuscript. F.Z. designed and synthesized the ligand TC114. Y.W. helped construct design, MD simulation, data analysis and assisted with manuscript preparation. J.Y. performed functional characterization of SMO mutants and TC114 ligand. G.W.H. performed structure determination and refinement. S.Z. helped with data analysis and assisted with manuscript preparation. A.I. helped to prepare crystal samples and collect diffraction data at LCLS. L.Y. synthesized a series of intermediate compounds. X.L. assisted with ligand screening and large-scale protein purification for crystallization. K.D. assisted with MD simulation analysis. V.D. and P.R.G. helped with the HDX experiments. C.G. processed XFEL data. G.N. operated LCP injector at LCLS. M.S.H. controlled SFX data collection at LCLS. M.A.H. helped with synchrotron data processing. V.C. supervised SFX data collection and structure determination at LCLS, and assisted with manuscript preparation. R.C.S. mentored X.Z. and participated in paper writing. W.T. supervised the functional characterization of SMO mutants and TC114 ligand. H.T. oversaw ligand design and synthesis, and participated in paper writing. F.X. conceived the project, supervised the research and wrote the manuscript.

## Additional information

**Competing interests:** R.C.S. and F.X. are founders and R.C.S. is a board member of Bird Rock Bio, a company focused on GPCR therapeutic antibodies. The remaining authors declare no competing financial interests.

