## [Peer review file · Nature Communications]

Reviewer #1 (Remarks to the Author):

NCOMMS-16-21344

This manuscript reports on the structure of Smoothed (Smo) bound to a stabilizing inhibitor (TC114) acquired both at room temperature using an XFEL at low temperatures using cryo-cooled crystals and a synchrotron source. The protein includes the 7TMD, the CRD and the connecting HD of Smo. Structures of the isolated Smo 7TMD bound to various antagonists and agonists have been reported previously (PMIDs 23636324, 25008467). Structures of both the liganded and unliganded forms of the isolated CRD (pmid 27437577, 24351982, 27545348) have also been solved. Finally, a recent report (pmid 27437577) solved structures of Smo, including the CRD, 7TMD and HD, bound to the CRD-agonist cholesterol or to the TMD-antagonist vismodegib. The TC114-Smo structure closely resembles the Vismodegib-bound structure and shows significant conformational differences compared to the (presumably active) cholesterol-bound structure, discussed by these authors in Figure 3.

Despite the fact that closely related structures have been recently published, I believe this paper has the potential to both confirm and extend the previous structural studies of Smo, but will require significant additional work. My main concern is that the paper has almost no functional tests of the structural model using structure-guided mutagenesis to assess effects on Hedgehog signaling. Mutational analysis to test structural predictions was an integral part of the recent comparable papers describing Smo structures (pmids 27545348 and 27437577) and is essential to maximizing the impact of this study. In addition, there are several other issues that need to be addressed or clarified:

Major Comments:

1. The effect of the E194M mutation (present in their crystallization construct) on the signaling activity of Smo should be assessed.
2. A major source of novelty in the paper is the design of a super-stabilizing SMO ligand, and the discussion that this structure could lead to new drugs to combat Hh cancers. It would be important to test whether TC114 can also block signaling by drug resistant mutants of SMO, like D473H, which is resistant to vismodegib.
2. In their discussion of the comparison between the cholesterol-bound, vismo-bound and TC114-bound structures, they describe a role for ECL3, including the glycan modification and I496, in occluding the sterol-binding site in the CRD. This leads the authors to suggest that the "outward tilt of helix VI and ECL3, and the displacement of the HD are likely to trigger a conformational change that activates the receptor." Since the mechanisms of SMO activation in response to sterols or Shh are still unclear, testing these models would be very valuable. For instance, the authors could make mutations that remove the glycan, or mutate I496, or even simply truncate the portion of ECL3 that is supposed to occlude the sterol-binding groove and ask what the effect is on signaling by SMO at baseline or in response to Shh, oxysterols, or cholesterol. If ECL3-occlusion is necessary for stabilizing the inactive state one might expect these mutations to be activating.

3. The normal-mode analysis and the HDX studies seem to be contradictory. The former suggests that the ECL3 and HD have significant flexibility but there is not much deuterium exchange seen in these regions in the HDX experiment. Since cholesterol has recently been shown to be a CRD agonist, it would be important compare the effect of 20S-OHC to cholesterol. HDX represents a unique approach, one the authors' could use more extensively to characterize conformational changes in SMO in response to ligands. One major issue is that the SMO protein used for HDX analysis includes a heterologous Rubredoxin protein inserted into ICL3, which will likely change the Smo conformational ensemble probed by HDX, especially on the cytoplasmic face (indeed exactly where the authors find most of their HDX effects). HDX experiments should be repeated on a SMO protein having native, unperturbed ICLs (or the fusion protein used should be evaluated to ensure that it displays intact Hedgehog signaling activity using a cell-based assay).

4. Related to the point #3 above the normal-mode analysis is likely to be unreliable because it does not account for the fact that SMO is in a membrane environment. Though computationally more intensive, the MD analysis should be redone simulating a membrane environment.

5. Two recent papers (PMIDs: 27705744 and 27545348) have suggested that cholesterol is a direct activating ligand for Smo, yet there is little discussion of this agonist effect. How might this impact the conformational changes proposed by the authors?

6. The authors suggest that the HD transmits the signal between the CRD and the 7TMD. The basis for this argument is not clear, nor is it tested with any mutagenesis experiments. The paper by Byrne et. al. shows mutagenesis data that argues against this hypothesis: mutations in the CRD itself (at the interface between the CRD and the rest of the molecule) leads to constitutive activity. Also, mutations of the disulfide bond in the HD domain (that should disrupt the HD conformation) lead to constitutive activity. The authors mistakenly state that this disulfide mutant cannot be activated; Byrne et. al. show that this mutant can be activated by the native ligand Shh.

Minor comments

1. The authors need to be much more careful in their efforts at citing the literature. Several papers that are highly relevant to this study and used in this study are not cited. Most important are papers showing that oxysterol/cholesterol bind to the CRD and describing the structures of the SMO CRD: PMIDs: 23954590, 27437577, 24351982, and 27705744. As one example amongst many, the structure of the zebrafish SMO CRD (from PMID 27437577) was used by the authors to solve the synchrotron structure, but this paper is never cited. These studies should be cited at the appropriate places throughout the manuscript.

2. The negative-stain EM is of poor quality and does not add anything to the paper. I would recommend that it be removed.

3. The authors suggest that the XEFL structure is of higher quality however there is no clear

evidence for this—the statistical table shows that they are essentially identical. Indeed both structures were solved in the same space group and appear very similar. I don't think the authors can conclude much about conformational changes based on a qualitative comparison of these structures and this discussion should be made much more conservative.

4. The comparison with the frizzled protein is speculative and does not add much to the story. The hinge domain is one of the least conserved regions of the Frizzled family.

Reviewer #2 (Remarks to the Author):

Review of "Crystal structure of a multi-domain human smoothed receptor in complex with a super stabilizing ligand"

Findings/claims in the manuscript:

1. Identification of a novel small molecule that thermostabilizes SMO for the purpose of facilitating its crystallization
2. Multidomain SMO crystals are obtained, and two data collection methods are used and compared. The crystal structure was determined at 2.9Å resolution.
3. Hydrogen-deuterium exchange is used to reveal regions of SMO that change their solvent accessibility upon ligand binding. These data provide evidence that the TMD has the potential to undergo conformational changes, despite the perfect overlap of the TMDs in various crystal structures (with several ligands bound).
4. The authors propose a model where a signal is transmitted from SMO CRD to TMD region through the hinge region, ECL3 and TM VI.
5. The authors compare SMO to other GPCRs including those of the Frizzled class and class B Glucagon receptor, which would be of interest to a broader audience.

The paper is generally clear and experiments well-executed. The only issue is the novelty of the findings. See below for more detailed comments and suggestions to improve the clarity of the manuscript.

General comments/suggestions:

1. Overall the manuscript is well written, however, the order of figures is confusing. It would be better to introduce the development and rationale behind the TC114 compound before presenting the SMO structure, which is bound to the molecule.
2. The SMO structural data are unfortunately limited in their novelty due to the recent publication of a similar structure (Byrne et al. 2016. Nature). However, this is undoubtedly an independent study and the authors have acknowledged and compared their findings to those in the literature.
3. The authors provide a comparison of the structure of their CRD region to refute claims by Huang et al (Cell, 2016) that the conformational changes in the CRD upon ligand binding are sufficient to activate SMO. It appears the presence of the hinge domain and ECL3 stabilize the CRD, limiting conformational changes upon ligand binding. A similar point is made by Luchetti et al. (ELife, 2016), which was published a few days after submission of this manuscript to Nature Communications. The authors might consider discussing this recent publication in support of their observation. This is an important point when considering a model for how a putative ligand might activate/inactivate SMO.

4. In the abstract the authors state: “we demonstrate that transmembrane helix VI, extracellular loop 3 (ECL3) and the HD play a central role in transmitting the signal from CRD to TMD employing a unique GPCR activation mechanism”. This is a slight exaggeration. I agree that they show the CRD has a differing position when bound to different ligands, that the hinge domain is flexible and that different ligands impact HDX of SMO regions, particularly in the cytoplasmic side of the TMD. However, as they point out, the direct observation of TMD conformation upon ligand binding remains elusive. I think their data support the idea, but do not directly demonstrate it and so this should be toned down.

The authors provide sufficient detail in methods and descriptions for other researchers to be able to reproduce this work. Moreover, the data are in line with that in other publications.

Minor comments:

- Page 3 Gli should be GLI
- It would be interesting to know why the authors chose to co-express SMO-FLA with vismodegib and not the TC114 compound (is it less stable?)
- Page 9 – Nature communications is a broad audience journal and so the authors should provide more explanation of their normal mode analysis for non-experts to interpret their conclusions – What is normal mode analysis? what would be expected? What has been tested? What assumptions are made?
- Similarly, the HDX analysis might benefit from further clarity such as explaining that HDX occurs in solvent accessible parts of the protein, so changes in conformation upon ligand binding that reveal or mask protein regions can be measured.
- In support of the notion that TMD conformational changes are critical for SMO activation are oncogenic mutations within transmembrane helices, particularly on the cytoplasmic side e.g. W535L – do these studies reveal anything about how these mutations might activate SMO?
- Page 11 “....while the CRD is *a* relatively..”
- Page 13: Citation(s) should be provided in the discussion about drug resistance/ mutations in the TMD region
- What are the technical reasons for why Byrne et al identified bound cholesterol and the authors did not?
 - Figure 3A: On the right side of the CRD there is a pink glycan (N188?) – it is not clear which chain it is associated with (presumably pink), therefore the asparagine side chain should be shown.

Reviewer #3 (Remarks to the Author):

This manuscript describes the X-ray crystal structure of the Smoothed receptor, and while the structure of this multi domain receptor has recently been published by the Siebold laboratory, this current study provides new mechanistic details that complements that published work. Overall this is a well written and interesting story using a combination of X-ray crystallography, biophysical analysis, and molecular dynamics to characterise this important GPCR receptor. They also describe a novel SMO ligand TC114 that allowed for stabilisation of the receptor structure. The authors clearly describe the differences between their structure and the Siebold lab structures in the results of the manuscript, however, it the Siebold structures are something that should be mentioned in the introduction of the manuscript as well.

There are a number of corrections that may strengthen and clarify points within the manuscript. With these changes I am supportive of eventual publication in Nature Communications.

Major points

1. The authors discuss the changed orientation of the CRD domain in the their two structures (Fig. 1b), as well as comparing their orientation to the published structures from the Siebold lab (Fig. 3A). From these structures it is very difficult to clearly observe the tilt in the CRD. This is particularly challenging in Fig. 3A, if the authors could more clearly label how each CRD in the different structures is tilted compared to their XFEL structure it might be more clear.

2. How do the HDX-MS decreases seen on the cytoplasmic face compare to previous studies using HDX-MS to examine ligand bound GPCR complexes?

3. The HDX-MS data is shown fitted to a curve in Fig. S7. These fits appear to be very imprecise, and there is no discussion of how these fits were generated. I would recommend completely removing these lines.

4. The authors discuss that the HD and ECL3 would likely act as a hinge between the CRD and TMD. HDX in this study is only used to compare the apo and TC114 bound states. The authors do not discuss the H/D exchange rates for this region compared to the stable CRD and TMD domains. It would be expected that these hinge regions would be much more dynamic and the secondary structure elements here would undergo much more rapid deuteration. This needs to be discussed in the text, as the authors describe in the abstract that

By combining the structural data, computer modelling, and hydrogen-deuterium-exchange analysis, we demonstrate that transmembrane helix VI, extracellular loop 3 (ECL3) and the HD play a central role in transmitting the signal from CRD to TMD employing a unique GPCR activation mechanism,

I am not sure how the TC114 H/D exchange data demonstrates the role of the ECL3 and HD transmitting this signal from CRD to TMD. This could be addressed either by a new figure in the supplement showing that indeed the hinge region is more flexible than would be expected.

Minor points

- In Fig 1A the primary sequence domain organisation is shown at a non-linear scale, and is confusing that the 30 amino acid hinge domain is the same length as the >300 amino acid TMD.

Responses are provided in blue text below each individual comment.

Reviewer #1:

My main concern is that the paper has almost no functional tests of the structural model using structure-guided mutagenesis to assess effects on Hedgehog signaling.

Response:

We have subsequently conducted a panel of cell-based functional tests of the structural model with structure-guided mutants to assess effects on Hedgehog signaling. The SMO cell assays are very challenging due to the usually low signals with or without ligand stimulation, especially for mutant tests. We spent significant efforts to set up this assay system and have the key mutants or treatment investigated. Details of these results have been added to the manuscript on page 6, 7, 10, 12. We have also included additional authors to the paper to reflect the contributions of this new work.

Major comments:

1. The effect of the E194M mutation (present in their crystallization construct) on the signaling activity of Smo should be assessed.

Response:

We tested the E194M mutant in a cell-based luciferase reporter assay, and compared the mutant with WT Smo. The results indicate that the E194M mutation elicits enhanced receptor activity to the WT. This result is mentioned on page 7 in the revised manuscript, and reported in Supplementary Fig. 2b.

2. A major source of novelty in the paper is the design of a super-stabilizing SMO ligand, and the discussion that this structure could lead to new drugs to combat Hh cancers. It would be important to test whether TC114 can also block signaling by drug resistant mutants of SMO, like D473H, which is resistant to vismodegib.

Response:

In the original manuscript, we tested the antagonist activity of TC114 and showed that its IC_{50} is comparable to its prototype LY2940680. In the revised manuscript, we tested TC114 antagonist activity on the SMO drug resistant mutant D473H and W535L. We showed that TC114 could block agonist SAG-induced signaling to these two mutants with an IC_{50} ~4-fold higher than WT, by a luciferase reporter assay. These results are mentioned on page 6 in the revised manuscript and reported in Supplementary Fig. 2a.

3. In their discussion of the comparison between the cholesterol-bound, vismo-bound and TC114-bound structures, they describe a role for ECL3, including the glycan modification and I496, in occluding the sterol-binding site in the CRD. This leads the authors to suggest that the “outward tilt of helix VI and ECL3, and the displacement of the HD are likely to trigger a conformational change that activates the receptor.” Since the mechanisms of SMO activation in response to sterols or Shh are still unclear, testing these models would be very valuable. For instance, the authors could make mutations that remove the glycan, or mutate I496, or even simply truncate the portion of ECL3 that is supposed to occlude the sterol-binding groove and ask what the effect is on signaling by

SMO at baseline or in response to Shh, oxysterols, or cholesterol. If ECL3-occlusion is necessary for stabilizing the inactive state one might expect these mutations to be activating.

Response:

In order to test our structure-based activation model, we made a panel of point mutations on the HD, ECL3 as well as glycan-modification residue. We characterized their effects on signaling by SMO at baseline conditions as well as in response to Shh and oxysterols. This data support our model that ECL3-occlusion is necessary for stabilizing the inactive state, as point mutations N493Q and I496R both showed increased basal activity of SMO signaling. Point mutations on the HD, V198R and K204A, can block the 20(S)-OHC, but not Shh, induced signaling. These results are described on page 10 and 12 in the revised manuscript and reported in Supplementary Fig. 2.

4. The normal-mode analysis and the HDX studies seem to be contradictory. The former suggests that the ECL3 and HD have significant flexibility but there is not much deuterium exchange seen in these regions in the HDX experiment. Since cholesterol has recently been shown to be a CRD agonist, it would be important compare the effect of 20S-OHC to cholesterol. HDX represents a unique approach, one the authors' could use more extensively to characterize conformational changes in SMO in response to ligands. One major issue is that the SMO protein used for HDX analysis includes a heterologous Rubredoxin protein inserted into ICL3, which will likely change the Smo conformational ensemble probed by HDX, especially on the cytoplasmic face (indeed exactly where the authors find most of their HDX effects). HDX experiments should be repeated on a SMO protein having native, unperturbed ICLs (or the fusion protein used should be evaluated to ensure that it displays intact Hedgehog signaling activity using a cell-based assay).

Response:

After an internal discussion, we agree that normal-mode analysis is quite preliminary and not appropriately suited for analyzing our model. We therefore deleted the normal-mode analysis part from the manuscript. As for the HDX analysis, we agree that such experiments with the non-fusion SMO protein would be more valuable and, hence, we tried very hard to conduct the experiment with the SMO protein that has native, unperturbed intracellular loops. However, the non-fusion SMO protein shows extremely poor expression (<0.05mg/L on yield) and reduced protein stability, compared to the Rubredoxin-fusion construct that we used for original HDX characterization. The poor stability of this protein makes it challenging to characterize with HDX, as the sequence coverage by MS is low. While we are continuing to optimize the HDX conditions, we think additional studies will be required to draw a comprehensive conclusion, which will be the focus of follow-up work. In parallel, we tested the effect of 20(S)-OHC treatment on the HDX results, and found no difference on the H/D exchange rate, likely due to some non-specific interaction between 20(S)-OHC and the detergent micelles.

Based on these experimental trials, we propose to maintain the original HDX results and conclusions in the manuscript. TC114 showed very clear protection in the indicated cytoplasmic region and we have confidence reporting this from the current data/results.

5. Related to the point #3 (should be #4) above the normal-mode analysis is likely to be unreliable because it does not account for the fact that SMO is in a membrane environment. Though computationally more intensive, the MD analysis should be redone simulating a membrane environment.

Response:

Since the normal-mode analysis does not represent SMO in a membrane environment, we have replaced it with a long 1- μ s MD simulation analysis with the protein embedded in a membrane environment. This analysis shows the overall stable conformation of the SMO CRD in the presence of cholesterol, similar to the 100-ns MD result reported in the paper by Byrne et al. Interestingly, the MD result showed that the cholesterol-stabilized CRD leans toward the membrane plane, in agreement with our structural comparison shown in Fig. 3a. This result is also consistent with our HDX analysis, where we do not see much H/D exchange at the CRD and TMD regions. We conclude that further investigation with a more intensive and thorough MD analysis is required, which will become the central focus of a follow-up paper. The result of suggested MD analysis is described on page 10 in the revised manuscript and reported in Supplementary Fig. 6.

6. Two recent papers (PMIDs: 27705744 and 27545348) have suggested that cholesterol is a direct activating ligand for Smo, yet there is little discussion of this agonist effect. How might this impact the conformational changes proposed by the authors?

Response:

According to the two recent papers (PMIDs: 27705744 and 27545348) and previous papers discussing SMO activation mechanisms, cholesterol and its oxy-metabolite, as well as cyclopamine adopt similar binding poses via a common 3- β hydroxyl mediated hydrogen bond network. Their agonistic functions for SMO were also indicated. In our cell based luciferase reporter assay, we stimulated the cells with 20(S)-OHC for its better solubility to circumvent the use of cyclodextrin (Supplementary Fig. 2c). Similarly, 20(S)-OHC seemed to be better for HDX experiments that required a very high concentration of ligand. In discussing our activation model, we mentioned both oxysterol (20(S)-OHC) and cholesterol in the revised manuscript.

7. The authors suggest that the HD transmits the signal between the CRD and the 7TMD. The basis for this argument is not clear, nor is it tested with any mutagenesis experiments. The paper by Byrne et. al. shows mutagenesis data that argues against this hypothesis: mutations in the CRD itself (at the interface between the CRD and the rest of the molecule) leads to constitutive activity. Also, mutations of the disulfide bond in the HD domain (that should disrupt the HD conformation) lead to constitutive activity. The authors mistakenly state that this disulfide mutant cannot be activated; Byrne et. al. show that this mutant can be activated by the native ligand Shh.

Response:

We thank the reviewer for pointing out this mistake. We have removed this statement on page 12 in the revised manuscript. We have included and discuss our own HD mutation results in the revised manuscript and reported this data in Supplementary Fig. 2.

Minor comments

1. The authors need to be much more careful in their efforts at citing the literature. Several papers that are highly relevant to this study and used in this study are not cited. Most important are papers showing that oxysterol/cholesterol bind to the CRD and describing the structures of the SMO CRD: PMIDs: 23954590, 27437577, 24351982, and 27705744. As one example amongst many, the structure of the zebrafish SMO CRD (from PMID 27437577) was used by the authors to solve the synchrotron structure, but this paper is never cited. These studies should be cited at the appropriate places throughout the manuscript.

Response:

We appreciate the reviewer's kind reminder and have included the key citations that the reviewer suggested at the appropriate places in the revised manuscript. We have also carefully checked all the citations throughout the paper to make sure the references are appropriate and reflect the state of the field.

2. The negative-stain EM is of poor quality and does not add anything to the paper. I would recommend that it be removed.

Response:

The EM data has been removed from the revised manuscript.

3. The authors suggest that the XEFL structure is of higher quality however there is no clear evidence for this—the statistical table shows that they are essentially identical. Indeed both structures were solved in the same space group and appear very similar. I don't think the authors can conclude much about conformational changes based on a qualitative comparison of these structures and this discussion should be made much more conservative.

Response:

We have revised the comparison between the two structures to be more conservative and concise. We have removed the comparison between the XFEL and Synchrotron structures, and have focused the discussion on only the XFEL structure in the revised manuscript.

4. The comparison with the frizzled protein is speculative and does not add much to the story. The hinge domain is one of the least conserved regions of the Frizzled family.

Response:

We believe these results are worth reporting. We agree that the hinge domain is not well conserved among the Frizzled family. In our revised manuscript, we deleted two ambiguous sentences and made a more cautious comparison on the CRD part, with a focus on the two ligand binding pockets, between SMO and the Frizzled proteins which are the most conserved in the Frizzled GPCR family.

Reviewer #2:

General comments/suggestions:

1. Overall the manuscript is well written, however, the order of figures is confusing. It would be better to introduce the development and rationale behind the TC114 compound before presenting the SMO structure, which is bound to the molecule.

Response:

We appreciate the reviewer's comment and have changed the order of Figures 1 & 2 as well as related text in the manuscript to present the design of TC114 first followed by presentation of the overall SMO structure.

2. The SMO structural data are unfortunately limited in their novelty due to the recent publication of a similar structure (Byrne et al. 2016. Nature). However, this is undoubtedly an independent study and the authors have acknowledged and compared their findings to those in the literature.

Response:

We thank the reviewer for this comment. Our study is independent and we acknowledged and compared our structure with the reported findings in the literature. We believe our study helps to strengthen the scientific data about this important receptor and further helps to clarify the role of the receptor in signal transduction.

3. The authors provide a comparison of the structure of their CRD region to refute claims by Huang et al (Cell, 2016) that the conformational changes in the CRD upon ligand binding are sufficient to activate SMO. It appears the presence of the hinge domain and ECL3 stabilize the CRD, limiting conformational changes upon ligand binding. A similar point is made by Luchetti et al. (eLife, 2016), which was published a few days after submission of this manuscript to Nature Communications. The authors might consider discussing this recent publication in support of their observation. This is an important point when considering a model for how a putative ligand might activate/inactivate SMO.

Response:

We have reviewed the new eLife paper by Luchetti et al, and agree with their point on the limited CRD conformations upon ligand binding. Since cholesterol and 20(S)-OHC bind to the same site on SMO and induce the same functional activity, the discussion on 20(S)-OHC is also applicable for cholesterol. We have added an appropriate citation to their study on page 9.

4. In the abstract the authors state: "we demonstrate that transmembrane helix VI, extracellular loop 3 (ECL3) and the HD play a central role in transmitting the signal from CRD to TMD employing a unique GPCR activation mechanism". This is a slight exaggeration. I agree that they show the CRD has a differing position when bound to different ligands, that the hinge domain is flexible and that different ligands impact HDX of SMO regions, particularly in the cytoplasmic side of the TMD. However, as they point out, the direct observation of TMD conformation upon ligand binding remains elusive. I think their data support the idea, but do not directly demonstrate it and so this should be

toned down.

Response:

We have toned down the statement on TMD conformations when discussing the SMO activation mechanism on Page 10 & 12 as well as in the Abstract as suggested by the reviewer.

Minor comments:

- Page 3 Gli should be GLI

Response:

Corrected.

- It would be interesting to know why the authors chose to co-express SMO-FLA with vismodegib and not the TC114 compound (is it less stable?)

Response:

Co-expression with vismodegib increased the protein expression yield; we also tested other ligands including TC114, but none of them worked better than vismodegib. We believe one key factor is the nitro group in TC114 which is very reactive.

- Page 9 – Nature communications is a broad audience journal and so the authors should provide more explanation of their normal mode analysis for non-experts to interpret their conclusions – What is normal mode analysis? what would be expected? What has been tested? What assumptions are made?

Response:

We have removed the normal-mode analysis according to Reviewer #1's comment and included MD data in its place.

- Similarly, the HDX analysis might benefit from further clarity such as explaining that HDX occurs in solvent accessible parts of the protein, so changes in conformation upon ligand binding that reveal or mask protein regions can be measured.

Response:

As suggested, we have added a short clarification on the HDX experiment on Page 10 in the revised manuscript.

- In support of the notion that TMD conformational changes are critical for SMO activation are oncogenic mutations within transmembrane helices, particularly on the cytoplasmic side e.g. W535L – do these studies reveal anything about how these mutations might activate SMO?

Response:

We do not see substantial conformational changes in W535 or its neighboring residues among all solved SMO structures, including agonist- or antagonist-bound TMD structures, as well as the two-domain structures, so we cannot currently correlate an effect between oncogenic mutations and SMO activation based on structural observations.

- Page 11 “...while the CRD is *a* relatively.”

Response:

We have removed this sentence in response to reviewer #1, point #4 above.

- Page 13: Citation(s) should be provided in the discussion about drug resistance/ mutations in the TMD region.

Response:

The citation (PMID: 26614022) was added on page 14.

- What are the technical reasons for why Byrne et al identified bound cholesterol and the authors did not?

Response:

In our crystal structure, the side chain of I496 from ECL3 is positioned to extend into the cholesterol binding site and this precludes cholesterol or oxysterol binding to the CRD. We have stated this on Page 8 in the revised manuscript.

- Figure 3A: On the right side of the CRD there is a pink glycan (N188?) – it is not clear which chain it is associated with (presumably pink), therefore the asparagine side chain should be shown.

Response:

The side chain of N493 which contains the glycan modification is now shown in the revised Fig. 3A, 3B.

Reviewer #3:

Major points

1. The authors discuss the changed orientation of the CRD domain in the their two structures (Fig. 1b), as well as comparing their orientation to the published structures from the Siebold lab (Fig. 3A). From these structures it is very difficult to clearly observe the tilt in the CRD. This is particularly challenging in Fig. 3A, if the authors could more clearly label how each CRD in the different structures is tilted compared to their XFEL structure it might be more clear.

Response:

Fig. 3b has been revised in the manuscript to clearly show the CRD tilt (direction and angle) on the cholesterol-bound structure and Vismodegib-bound structure with respect to our XFEL structure. The synchrotron structure has been removed from this figure for clarity.

2. How do the HDX-MS decreases seen on the cytoplasmic face compare to previous studies using HDX-MS to examine ligand bound GPCR complexes?

Response:

The HDX-MS decreases observed on the cytoplasmic side in our SMO structure are comparable to previous studies on the ligand-bound or apo GCGR structure (PMID: 26227798, Fig. 2).

3. The HDX-MS data is shown fitted to a curve in Fig. S7. These fits appear to be very imprecise, and there is no discussion of how these fits were generated. I would recommend completely removing these lines.

Response:

The HDX-MS data curve fits were generated in prism (we added the sentence “data was fitted to a simple nonlinear regression (least squares) best fit model (X is log and Y is linear) using GraphPad Prism” in the revised manuscript on Page 20—the Methods session). If we had more time points between 0 and 10s (the earliest we could measure), then the curves would look much smoother. So we think these fits in our Fig. S5 are precise. We have removed these lines to make it appear more reasonable according to the reviewer's suggestion.

4. The authors discuss that the HD and ECL3 would likely act as a hinge between the CRD and TMD. HDX in this study is only used to compare the apo and TC114 bound states. The authors do not discuss the H/D exchange rates for this region compared to the stable CRD and TMD domains. It would be expected that these hinge regions would be much more dynamic and the secondary structure elements here would undergo much more rapid deuteration. This needs to be discussed in the text, as the authors describe in the abstract that “By combining the structural data, computer modelling, and hydrogen-deuterium-exchange analysis, we demonstrate that transmembrane helix VI, extracellular loop 3 (ECL3) and the HD play a central role in transmitting the signal from CRD to TMD employing a unique GPCR activation mechanism”, I am not sure how the TC114 H/D exchange data demonstrates the role of the ECL3 and HD transmitting this signal

from CRD to TMD. This could be addressed either by a new figure in the supplement showing that indeed the hinge region is more flexible than would be expected.

Response:

We conducted a panel of functional assays with SMO mutants that contain structure-guided point mutations on the HD and ECL3 domains, respectively. The results are now included in Supplementary Fig. 2 and described on page 10, 12 in the revised manuscript supporting the observation that both HD and ECL3 are critical for signaling. For the past 2 months, we have conducted additional HDX-MS experiments using the SMO protein without an ICL fusion protein to probe conformational flexibility in a more native state. However, as noted in the reply to reviewer 1, the non-fusion SMO protein is very unstable and hence not amenable for HDX analysis due to low MS sequence coverage. We therefore cannot draw conclusions from the HDX analysis regarding the flexibility of the HD domain; further investigation along this line is required and we are continuing this work in a series of follow-up studies. As such, we toned down the wording in the abstract to make the statement more precise and consistent.

Minor points

- In Fig 1A the primary sequence domain organization is shown at a non-linear scale, and it is confusing that the 30 amino acid hinge domain is the same length as the >300 amino acid TMD.

Response:

A new Fig. 2A (originally Fig. 1A and here moved to Fig. 2A as suggested by Reviewer#2) is made to reflect the length of each domain.

Reviewers' comments:

Reviewer #1 (Remarks to the Author):

The authors have done a remarkable job of addressing nearly all the concerns raised by the reviewers, either by removing data that was not convincing or by performing additional experiments. I recommend publication without additional modifications.

Reviewer #2 (Remarks to the Author):

Comments on revised version of “Crystal structure of a multi-domain human Smoothed receptor in complex with a super stabilizing ligand.”

The authors have reordered the manuscript for clarity, improved their citations and included functional analyses. However, a number of errors have been introduced upon revision and there are still problems with referencing.

Major comment:

In response to Reviewer #1 the authors now include functional assays. The authors highlight the challenge of working with these assays – although they are routinely used in several labs. Nevertheless, if poor or inconsistent responses are being observed with this assay it is key to include the right controls. Although I do not think it was the authors' intent that TC114 could be used as an alternative therapeutic option, the new data assessing its activity on vismodegib-resistant SMO mutants (supplementary fig. 2a) should include vismodegib as a control. Alternatively, the levels of receptor introduced into the reporter line should be established as this is clearly an important variable. Otherwise, the biological significance of the reported IC50 shift is meaningless. Additionally, to be precise, the SMO-W535L has reduced *sensitivity* to vismodegib, however, it has not been shown to cause resistance in patients and in fact oncogenic SMO mutant-driven tumors appear to respond to the drug (PMID: 25759019).

Minor comments:

1. Incorrect statements:

Page 12: Lack of a known endogenous ligand for SMO.

PMIDs 27705744 and 27545348 show that cholesterol is an endogenous ligand/cofactor that binds the SMO CRD.

Page 13: “The only FDA approved drug...”. On the contrary, LDE-225/sonidegib has also been approved by the FDA (see e.g. PMID: 27189494)

2. Previous comment and response:

Citation(s) should be provided in the discussion about drug resistance/ mutations in the TMD region.

Response: The citation (PMID: 26614022) was added on page 14. **

The recent primary literature should be cited rather than a review: i.e. PMIDs: 25759020 and 25759019 (pages 13 and 14)

3. Order of figures:

Supp. Fig. 1b is mentioned before 1a.

Supp Fig. 2c is mentioned after Supp Figs 3 and 4 and the data is only reported in the discussion.

4. As noted in point 3, the authors perform assays on V198 and K204 mutants in Supp. Fig. 2c. In addition to explaining why these assays were done and reporting them in the results, the residues should be highlighted on the crystal structure.

In PMID: 27705744 a similar result (mutants that disrupted 20 (S) OHC activity but maintained SHH sensitivity) was reported. The interpretation was that 20 (S) OHC activates SMO in a distinct way compared to endogenous signalling because it binds differently than cholesterol – could this also explain the V198 and K204 mutant data?

5. Vismodegib should be vismodegib (i.e not capitalized).

6. Page 13: "Long-term administration, however, could lead to the development of resistance toward this drug²⁵". Replace "could" with "can". There are several studies reporting resistance to the drug in BCC as well as in medulloblastoma.

7. Discussion is repetitive e.g. Para 1: "Since no activation-related conformational changes in the TMD have been seen in any reported SMO structures"

Para 2: "no TMD conformational changes have yet been observed in any SMO structure"

These points should be made in a more concise manner.

8. General typos/grammar:

Supp fig. 6 legend: "blue line indicates ~7° twist angle of SMO-LT114"

Page 3: "Yes, the mechanism"

Page 9: "we do not see large-degree", "higher flexibility on the CRD"

9. Page 12: "CRD exerts allosteric modulation of the TMD, and vice-versa". Needs a reference.

10. Previous comment:

What are the technical reasons for why Byrne et al identified bound cholesterol and the authors did not?

Response: In our crystal structure, the side chain of I496 from ECL3 is positioned to extend into the cholesterol binding site and this precludes cholesterol or oxysterol binding to the CRD. We have stated this on Page 8 in the revised manuscript.

On page 8 of the revised manuscript: "In fact, our structure demonstrates that in the absence of sterol binding, ECL3 interacts with the CRD hydrophobic groove to stabilize the CRD in a "closed" conformation (Fig. 2c)."

This section/description could be improved with more detail about how the structure led to the functional assays– why pick only N493 and I496 to mutate?

The interaction of ECL3 and the CRD hydrophobic groove is interesting – does the same happen in the vismodegib-bound structure from PMID: 27437577? Is this a general influence of TMD inhibitors on the CRD pocket/a true "closed conformation"? or is ECL3 just flexible. By contrast, Byrne et al show I496 forms part of the sterol binding pocket – so this could certainly be a residue that is displaced upon cholesterol binding – a potential discussion point.

Reviewer #3 (Remarks to the Author):

The authors have successfully addressed all of my concerns. I support publication in its current form.

Responses are provided in blue text below each individual comment.

Reviewer #2

Major comment:

In response to Reviewer #1 the authors now include functional assays. The authors highlight the challenge of working with these assays – although they are routinely used in several labs. Nevertheless, if poor or inconsistent responses are being observed with this assay it is key to include the right controls. Although I do not think it was the authors' intent that TC114 could be used as an alternative therapeutic option, the new data assessing its activity on vismodegib-resistant SMO mutants (supplementary fig. 2a) should include vismodegib as a control. Alternatively, the levels of receptor introduced into the reporter line should be established as this is clearly an important variable. Otherwise, the biological significance of the reported IC50 shift is meaningless. Additionally, to be precise, the SMO-W535L has reduced sensitivity to vismodegib, however, it has not been shown to cause resistance in patients and in fact oncogenic SMO mutant-driven tumors appear to respond to the drug (PMID: 25759019).

Thanks to the helpful comments, we have made below revision:

- (1) We agree that W535L is not a drug resistant mutation and we have removed W535L data from Supp. Fig. 2.
- (2) To clarify, this paper is not intended to be about drug discovery. The tool ligand TC114 we developed is intended as a powerful utility chemical probe and to be used in co-crystallization studies. We feel that the comparison data to the drug vismodegib would take the paper further down a path that is not the focus or intent. We discussed the rationale and goal of our compound design in the “Discussion” session on Page 13 to clarify this focus.
- (3) We agree that the receptor level could be a potential variable in the reported luciferase assay, though we have been very careful with the experiment by using the same number of cells, plasmids, and transfection reagent for each transfection. Following the reviewer's suggestion, we have checked the target protein expression level for each SMO mutant relative to WT by western blot analysis, and confirmed that these single-point mutations do not significantly alter the level of receptor introduced into the reporter line (see revised Supp. Fig. 2&8). The western blot analysis has been added to the “Method” session.

Minor comments:

1. Incorrect statements:

Page 12: Lack of a known endogenous ligand for SMO.

PMIDs 27705744 and 27545348 show that cholesterol is an endogenous ligand/cofactor that binds the SMO CRD.

Page 13: “The only FDA approved drug...”. On the contrary, LDE-225/sonidegib has also been approved by the FDA (see e.g. PMID: 27189494)

Response:

We thank the reviewer for this comment and have corrected the statements as mentioned on pages 12 and 13.

2. Previous comment and response:

Citation(s) should be provided in the discussion about drug resistance/ mutations in the TMD region.

The recent primary literature should be cited rather than a review: i.e. PMIDs: 25759020 and 25759019 (pages 13 and 14)

Response:

The citations (PMID: 25759020 and 25759019) have been added on page 13.

3. Order of figures:

Supp. Fig. 1b is mentioned before 1a.

Supp Fig. 2c is mentioned after Supp Figs 3 and 4 and the data is only reported in the discussion.

Response:

We have changed the order of Supplementary Fig. 1a and 1b. We have moved Supp. Fig. 2c to a new Supp. Fig. 8b.

4. As noted in point 3, the authors perform assays on V198 and K204 mutants in Supp. Fig. 2c. In addition to explaining why these assays were done and reporting them in the results, the residues should be highlighted on the crystal structure.

In PMID: 27705744 a similar result (mutants that disrupted 20 (S) OHC activity but maintained SHH sensitivity) was reported. The interpretation was that 20 (S) OHC activates SMO in a distinct way compared to endogenous signalling because it binds differently than cholesterol – could this also explain the V198 and K204 mutant data?

Response:

We have made a new Supp. Fig. 8a to highlight the residues on the crystal structure. A short discussion concerning “20(S)-OHC activates SMO in a distinct way” with citation of PMID 27705744 was added on Page 11.

5. Vismodegib should be vismodegib (i.e not capitalized).

Response:

Corrected and checked throughout.

6. Page 13: "Long-term administration, however, could lead to the development of resistance toward this drug²⁵". Replace "could" with "can". There are several studies reporting resistance to the drug in BCC as well as in medulloblastoma.

Response:

Corrected.

7. Discussion is repetitive e.g. Para 1:” Since no activation-related conformational changes in the TMD have been seen in any reported SMO structures”

Para 2: “no TMD conformational changes have yet been observed in any SMO structure”

These points should be made in a more concise manner.

Response:

The first sentence on page 11 has been modified accordingly.

8. General typos/grammar:

Supp fig. 6 legend: “blue line indicates $\sim 7^\circ$ twist angle of SMO-LT114”

Page 3: “Yes, the mechanism”

Page 9: “we do not see large-degree”, “higher flexibility on the CRD”

Response:

Corrected.

9. Page 12: “CRD exerts allosteric modulation of the TMD, and vice-versa”. Needs a reference.

Response:

The citation (PMID: 23954590) was added on Page 12.

10. Previous comment:

What are the technical reasons for why Byrne et al identified bound cholesterol and the authors did not?

Response: In our crystal structure, the side chain of I496 from ECL3 is positioned to extend into the cholesterol binding site and this precludes cholesterol or oxysterol binding to the CRD. We have stated this on Page 8 in the revised manuscript.:

On page 8 of the revised manuscript: “In fact, our structure demonstrates that in the absence of sterol binding, ECL3 interacts with the CRD hydrophobic groove to stabilize the CRD in a “closed” conformation (Fig. 2c).”

This section/description could be improved with more detail about how the structure led to the functional assays– why pick only N493 and I496 to mutate?

Response:

We focused mutagenesis studies on the residues N493 and I496 from ECL3 in regulation of sterol binding site and CRD conformation based on our structure, and to complement the elegant work by Byrne et al. This explanation is on Page 7&9. We agree with Reviewer #2 that the differences and similarities are important and this is an area under further investigation that will include additional sites being studied. We want to be careful about expanding the level of detail beyond what we are confident in, and therefore believe we have found a good balance in the current manuscript.

The interaction of ECL3 and the CRD hydrophobic groove is interesting – does the same happen in the vismodegib-bound structure from PMID: 27437577? Is this a general influence of TMD inhibitors on the CRD pocket/a true “closed conformation”? or is ECL3 just flexible. By contrast, Byrne et al show I496 forms part of the sterol binding pocket – so this could certainly be a residue that is displaced upon cholesterol binding – a potential discussion point.

Response:

We used the term “open” or “closed” conformation for CRD from the SMO CRD structure paper (*Cell*, PMID: 27545348). We described our observation based on a comparison with their CRD structures, and state that our structure showed a “closed conformation” for CRD in the absence of sterol binding. The ECL3 loop is partially disordered in the vismodegib-bound structure, while in our structure, we observe that this loop leans to CRD and stabilizes its conformation. As noted in the earlier comment, we discuss I496 on Page 7, and prefer not to extend the discussion of its potential role without further data to support.

REVIEWERS' COMMENTS:

Reviewer #2 (Remarks to the Author):

I am satisfied with the changes that the authors have made to address my comments and believe the manuscript should now be accepted.